# Structural Characterization and Assessment of Anti-Inflammatory and Anti-Tyrosinase Activities of Polyphenols from *Melastoma normale*

**DOI:** 10.3390/molecules26133913

**Published:** 2021-06-26

**Authors:** Rui-Jie He, Jun Li, Yong-Lin Huang, Ya-Feng Wang, Bing-Yuan Yang, Zhang-Bin Liu, Li Ge, Ke-Di Yang, Dian-Peng Li

**Affiliations:** 1School of Chemistry and Chemical Engineering, Guangxi University, Nanning 530004, China; hrj937@gxib.cn (R.-J.H.); geli_2009@163.com (L.G.); 2Guangxi Key Laboratory of Functional Phytochemicals Research and Utilization, Guangxi Institute of Botany, Guangxi Zhuang Autonomous Region and Chinese Academy of Sciences, Guilin 541006, China; wyf@gxib.cn (Y.-F.W.); yby@gxib.cn (B.-Y.Y.); lzb@gxib.cn (Z.-B.L.); 3State Key Laboratory for the Chemistry and Molecular Engineering of Medicinal Resources, College of Chemistry and Pharmacy, Guangxi Normal University, Guilin 541004, China; lijun9593@gxnu.edu.cn

**Keywords:** *Melastoma normale*, *Melastoma*, polyphenols, tyrosinase, anti-inflammatory, ellagitannins

## Abstract

Polyphenols, widely distributed in the genus *Melastoma* plants, possess extensive cellular protective effects such as anti-inflammatory, anti-tyrosinase, and anti-obesity, which makes it a potential anti-inflammatory drug or enzyme inhibitor. Therefore, the aim of this study is to screen for the anti-inflammatory and enzyme inhibitory activities of compounds from title plant. Using silica gel, MCI, ODS C18, and Sephadex LH-20 column chromatography, as well as semipreparative HPLC, the extract of *Melastoma normale* roots was separated. Four new ellagitannins, Whiskey tannin C (**1**), 1-*O*-(4-methoxygalloyl)-6-*O*-galloyl-2,3-*O*-(*S*)-hexahydroxydiphenoyl-*β*-d-glucose (**2**), 1-*O*-galloyl-6-*O*-(3-methoxygalloyl)-2,3-*O*-(*S*)-hexahydroxydiphenoyl-*β*-d-glucose (**3**), and 1-*O*-galloyl-6-*O*-vanilloyl-2,3-*O*-(*S*)-hexahydroxydiphenoyl-*β*-d-glucose (**4**), along with eight known polyphenols were firstly obtained from this plant. The structures of all isolates were elucidated by HRMS, NMR, and CD analyses. Using lipopolysaccharide (LPS)-stimulated RAW2 64.7 cells, we investigated the anti-inflammatory activities of compounds **1**–**4**, unfortunately, none of them exhibit inhibit nitric oxide (NO) production, their IC_50_ values are all > 50 μM. Anti-tyrosinase activity assays was done by tyrosinase inhibition activity screening model. Compound **1** showed weak tyrosinase inhibitory activity with IC_50_ values of 426.02 ± 11.31 μM. Compounds **2**–**4** displayed moderate tyrosinase inhibitory activities with IC_50_ values in the range of 124.74 ± 3.12–241.41 ± 6.23 μM. The structure–activity relationships indicate that hydroxylation at C-3′, C-4′, and C-3 in the flavones were key to their anti-tyrosinase activities. The successful isolation and structure identification of ellagitannin provide materials for the screening of anti-inflammatory drugs and enzyme inhibitors, and also contribute to the development and utilization of *M. normale*.

## 1. Introduction

Inflammation, as a common clinical pathological process, is closely related to many diseases such as arthritis, psychosis, cardiovascular and cerebrovascular diseases, and cancer [1]. Tyrosinase is the key enzyme of melanin synthesis, and its overexpression can lead to pigmentation diseases such as freckles, chloasma and melanoma [2]. At present, anti-inflammatory drugs such as glucocorticoids, insulin, and tyrosinase inhibitors such as kojic acid have been proven to have significant side effects [3,4]. Therefore, finding out the effective and lower side effects anti-inflammatory drugs and tyrosinase inhibitor is of great importance. Polyphenols, the characteristic component of *Melastoma* plants, have broad cytoprotective effects, such as anti-oxidation, anti-inflammatory, anti-tyrosinase [5], and anti-obesity [6,7], which make it a potential anti-inflammatory drug or enzyme inhibitor. *Melastoma normale* D. Don, a shrub of the Melastomaceae family, is widespread in Nepal, India, Myanmar, Malaysia, Philippines, and China [8]. The roots of *M. normale*, “Yang KaiKou” in Chinese, contain abundant polyphenols, which have been demonstrated as bioactive constituents corresponding to the anti-inflammatory effect of this plant [9,10,11]. With the aim to screen for the anti-inflammatory and enzyme inhibitory activities of compounds from the title plant, the roots of *M. normale* were extracted by 80% aqueous acetone and subsequent separated by various chromatographic methods to yield twelve polyphenols. The structures of the isolates were characterized by comprehensive spectroscopic data analyses. Moreover, the anti-inflammatory activities of new compounds **1**–**4** were investigated to develop Polyphenols as a novel anti-inflammatory drug. In addition, anti-tyrosinase activities of compounds **1**–**12** were evaluated, and the structure–activity relationships of compounds **7**–**12** were studied. In this study, there are five ellagitannins were isolated from *M. normale*. As we all known, ellagitannin is esters of hexahydroxydiphenoic acid and monosaccharide. The separation of these compounds is often difficulty due to its high polarity, easy to be adsorbed by separation materials, and unstable to light and heat. Moreover, because of its large molecular weight, many identical structural units in the structure, as well as the serious overlap of NMR, the determination of its structure has always been a difficult problem. Therefore, separation and structural identification of these compounds is still a difficult problem in the field of natural product chemistry. The successful isolation and structural identification of the ellagitannins provide materials for this experiment, and also contribute to the development and utilization of the plant.

## 2. Results and Discussion

The EtOAc fraction of the 80% aqueous acetone extract of *M. normale* roots was purified using various chromatographic methods to give four new compounds **1**–**4** and eight known compounds **5**–**12**. The structures of **1**–**12** are shown in Scheme 1. The known compounds **5**–**12** were identified as mongolicain A (**5**) [12], 1-hydroxy-3,4,5-trimethoxy phenyl-1-*O*-[6′-*O*-(4″-carboxy-1″,3″,5″-trihydroxy)phenyl]-*β*-d-glucopyranoside (**6**) [13], kaempferol (**7**) [14], kaempferol-3-rhamnoside (**8**) [15], gentisic acid-5-*O*-*β*-d-(6’-*O*-galloyl)-glucopyranside (**9**) [16], quercetin (**10**) [17], quercetin-3-*O*-*α*-L-rhamnoside (**11**) [18], and myricetin-3-*O*-*α*-L-rhamnopyranoside (**12**) [19], respectively. All compounds were firstly isolated from this plant. Moreover, the ellagitannins are firstly obtained from the roots of *M. normale* [10,11,20]. 

### 2.1. Structure Elucidation

Compound **1** was obtained as a yellow amorphous powder and showed the positive a coloration characteristic of ellagitannins with the NaNO_2_-AcOH reagent. The molecular formula C_45_H_34_O_27_ was established by the HRESIMS spectrum with a negative-ion peak at *m*/*z* [M − H]^−^ 1005.1240 (calcd 1005.1215). The ^1^H-NMR data (Table 1) revealed at least two hexahydroxydiphenoyl (HHDP) groups (*δ* 6.60, 6.70, and 6.83), six methine groups, three oxygen-bearing methylene groups, and two methyl groups. The ^13^C NMR data (Table 1) revealed six ester carbonyl groups at *δ* 170.5, 170.1, 168.9, 168.8, 167.6, and 163.3. The presence of two ethoxyls was indicated by the ^1^H-^1^H COSY correlations (Figure 1). The NMR data (Table 1) and ^1^H-^1^H COSY correlations (Figure 1) revealed a polyalchohol unit, which was similar to the open-chain glucose unit of vescalagin [21]. The relatively lowfield carbon chemical shifts of the glucose C-2–C-6 and HMBC correlations (Figure 1) indicated that hydroxyl groups at the glucose C-2–C-6 were acylated. One of the two HHDP groups attached to the glucose C-4 and C-6 was confirmed by a large difference the chemical shifts between the H-6a (*δ* 4.95) and H-6b (*δ* 3.95) [22], and HMBC correlations of the glucose H-4 and H-6 with the carbonyl carbons C_HHDP-7′_ (*δ* 168.9) and C_HHDP-7‴_ (*δ* 170.1), respectively. A cyclopentenone ring possessing an ethoxycarbonyl group could be constructed by analysis of the carbon signals due to a carbonyl (*δ* 202.9 (Cp-3)), an ester carbonyl (*δ* 170.5 (Cp-7)), two olefinic (*δ* 143.8 (Cp-4) and 158.2 (Cp-5)), a methine (*δ* 46.3 (Cp-1)), and an oxygen-bearing quaternary carbon (*δ* 84.9, Cp-2). This was supported by the HMBC correlations of methylene protons (*δ* 4.25) with Cp-7 and H_Cp_-1 (*δ* 5.46) with Cp-2, Cp-3, Cp-4, Cp-5, and Cp-7. In addition, the connection of the Cp-4 to the glucose C-1 was demonstrated by the HMBC correlations of the glucose H-1 with Cp-3, Cp-4, and Cp-5. The relatively high-field ester carbonyl carbon chemical shift at *δ* 163.3 (Cp-6) suggested that this carbonyl was linked to a double bond and form a *δ*-lactone ring with the glucose C-2 hydroxyl group. This was confirmed by HMBC correlation of the ester carbonyl carbon (*δ* 163.3) with glucose H-2. The correlations of H_Cp_-1 with C_HHDP-2_ (*δ* 125.0), C_HHDP-3_ (*δ* 111.2), and C_HHDP-4_ (*δ* 146.4) in the HMBC spectrum (Figure 1) indicated that Cp-1 was conjugated with C_HHDP-3_. Moreover, the glucose C-3 and C-5 hydroxyl groups formed other two lactone rings with two carbonyl groups attached to the hexahydroxybiphenyl moiety. This was demonstrated by the HMBC correlations of glucose H-3 with ester carbonyl carbon (C_HHDP-7_
*δ* 167.6) and the glucose H-5 and the aromatic proton H_HHDP-3′_ with ester carbonyl carbon (C_HHDP-7′_
*δ* 168.9). The key HMBC correlations of the methylene protons (*δ* 3.60 and 3.52) with the glucose C-1 (*δ* 68.0) and the glucose H-1 (*δ* 4.34) with the methylene carbon (*δ* 66.0) indicated that the ethoxyl moiety was located at the glucose C-1. Therefore, the 2D structure of **1** was established. This was supported by comparing the spectroscopic data of **1** with those of Whiskey tannin B [23]. The small coupling constant between glucose H-1 and H-2 (*J* < 2.0 Hz) revealed that the configuration at glucose C-1 of **1** was the same as that of vescalagin (*J* = 2.0 Hz) [21], and different from that of Whiskey tannin B (*J* = 6.4 Hz) [23]. Assuming that **1** was derived from vescalagin, construction of a Dreiding model of **1** indicated that the benzyl methine proton (Cp-1) of the cyclopentenone ring must be *β*-configuration, because its fused ring system, including two aromatic rings, a cyclopentene, and 6- and 10-membered lactone rings, was so rigid that an alternative model could not be constructed. The absence of ROE correlations between the Cp-1 proton and the methoxyl group suggested an α-orientation of the ethoxycarbonyl group. The CD spectrum of **1** (Appendix A) showed a positive Cotton effect at 240 nm and a negative one at 263 nm indicating that the atropisomerism of the HHDP group had the *S*-configuration [24]. Based on these results, the structure of compound **1**, named Whiskey tannin C, was established as depicted in Scheme 1.

Compound **2** was isolated as a brown amorphous powder. The molecular formula C_35_H_28_O_22_ was determined based on the [M − H]^−^ peak at *m*/*z* 799.0983 (calcd 799.0999) in its HRESIMS spectrum. The ^1^H NMR data (Table 2) revealed two galloyl groups (*δ*_H_ 7.12 (2H, s) and 7.11 (2H, s)), an HHDP group (*δ*_H_ 6.72 (1H, s) and 6.43 (1H, s)), and a methoxyl group (*δ*_H_ 3.87 (3H, s)). The relatively low-field chemical shifts and the relatively large coupling patterns of the sugar protons indicated that glucose cores in **2** adopt the ^4^C_1_ conformation and the acyl groups should be located at *O*-1, *O*-2, *O*-3, and *O*-6. The location of the HHDP group at glucose *O*-2/*O*-3 was supported by evidence from the HMBC correlations, the ^1^H-^1^H COSY correlations (Figure 2), as well as the chemical shift (*δ* 92.6) of the anomeric carbon signals [25]. The (*S*) configuration of the HHDP group was confirmed by the CD spectrum, which exhibited a positive cotton effect at 238 nm and a negative cotton effect at 264 nm [24]. The coupling constant of the anomeric proton in **2** was 8.5 Hz, indicating that the sugar moiety had a *β* configuration. Comparison of its NMR data (Table 2) with those of 1,6-di-*O*-galloyl-2,3-*O*-(*S*)-hexahydroxydiphenoyl-*β*-d-glucose [20] revealed that the hydroxy group at galloyl-C-4 in the latter was replaced by a methoxyl group in the former. The molecular formula, the chemical shifts of galloyl-C-3 (*δ*_C_ 151.5), galloyl-C-4 (*δ*_C_ 141.3), galloyl-C-5 (*δ*_C_ 151.5), and -OCH_3_ (*δ*_C_ 60.8), and the correlations of aromatic proton (*δ*_H_ 7.12) and methoxyl proton (*δ*_H_ 3.87) with galloyl-C-4 (*δ*_C_ 141.3) in the HMBC spectrum supported these changes in **2**. Acid hydrolysis and subsequent methylation and silylation of **2** established that the sugar in the molecule was D-glucose. The structure of **2** was therefore established as 1-*O*-(4-methoxygalloyl)-6-*O*-galloyl-2,3-*O*-(*S*)-hexahydroxydiphenoyl-*β*-d-glucose.

Compound **3** had the same molecular formula as compound **2**, as determined by its HRESIMS data. Comparison between the NMR data of **3** and **2** (Table 2) revealed that the methoxyl group at galloyl-C-4 in **2** shifted to galloyl-C-3′ in **3**. This was confirmed by the HMBC correlations (Figure 2) of the methoxyl protons (*δ*_H_ 3.88) and aromatic proton (*δ*_H_ 7.24) with galloyl-C-3′ (*δ*_C_ 149.1) in **3**, and the chemical shifts of the methoxyl group, galloyl-C-3, galloyl-C-4, and galloyl-C-3′ that shifted from *δ*_C_ 60.8, 151.5, 141.3, and 146.5 in **2** to *δ*_C_ 56.7, 146.6, 140.7, and 149.1 in **3**, respectively. A positive cotton effect at 237 nm and a negative one at 262 nm in the CD spectrum indicated the (*S*) configuration of the HHDP group for **3 [24]**. The *β* configuration of the sugar moiety was deduced from the coupling constant (*J* = 8.5 Hz) of the anomeric proton of the glucosyl moiety. Acid hydrolysis of **3** with 1 M HCl yielded D-glucose, which was confirmed by TLC and GC analyses. Hence, compound **3** was identified as 1-*O*-galloyl-6-*O*-(3-methoxygalloyl)-2,3-*O*-(*S*)-hexahydroxydiphenoyl-*β*-d-glucose.

The molecular formula of **4** was deduced from its HRESIMS spectrum (*m*/*z* 783.1032, [M − H]^−^) as C_35_ H_28_ O_21_. Its molecular weight was 16 mass units less than that of **3**, which may be attributed to the absence of a hydroxyl and the presence of an aromatic proton (*δ*_H_ 6.87) at vanilloyl-C-5′ in **4** (Table 2). This was confirmed from the presence of a 1,2,4-trisubstituted aromatic moiety (*δ*_H_ 7.60 (dd, *J* = 8.3, 1.8 Hz), 6.87 (d, *J* = 8.3 Hz), and 7.57 (d, *J* = 1.8 Hz)) and the chemical shift change of galloyl-C-5′ at *δ*_C_ 145.8 in **3** to vanilloyl-C-5′ at *δ*_C_ 115.9 in **4**. The methoxyl protons (*δ*_H_ 3.90) and aromatic proton (*δ*_H_ 6.87) correlations with vanilloyl-C-3′ in the HMBC spectrum (Figure 2) further confirmed the aromatic proton (*δ*_H_ 6.87) at vanilloyl-C-5′. The atropisomerism of the HHDP was shown to be an *S* configuration by appearance of positive and negative cotton effects at 239 nm and 263 nm, respectively [24]. The coupling constants of the anomeric proton in **4** were 8.5 Hz reminiscent of *β*-anomeric configuration. The acid hydrolysis revealed that **4** had the same sugar units as **3**. Thus, compound **4** was identified as 1-*O*-galloyl-6-*O*-vanilloyl-2,3-*O*-(*S*)-hexahydroxydiphenoyl-*β*-d-glucose.

### 2.2. Anti-Inflammatory Activity Assays

Four new compounds **1**–**4** were tested for potential anti-inflammatory activity by measuring the inhibition of the nitric oxide (NO) production. Unfortunately, none of them displayed significantly anti-inflammatory activity. The IC_50_ values for the inhibition of NO production by compounds **1**–**4** are all > 50 μM. It has been reported that the known flavonols kaempferol (**7**) [26], quercetin (**10**) [27], and myricetin-3-*O*-*α*-L–rhamnopyranoside (**12**) [28] have anti-inflammatory activity, but kaempferol-3-rhamnoside (**8**) [26] and quercetin-3-*O*-*α*-L-rhamnoside (**11**) [29] have no anti-inflammatory activity. From the experimental results and literature data, the flavonoids in the title plant have better anti-inflammatory activity than ellagitannins, it may be the active ingredient corresponding to the anti-inflammatory effect of this plant.

### 2.3. Tyrosinase Inhibitory Activity Assays

All compounds were investigated for potential tyrosinase inhibitory activity. As shown in Table 3, new compounds **2**–**4** displayed moderate tyrosinase inhibitory activities. New compound **1** exhibited weak tyrosinase inhibitory activity. Quercetin (**10**) [30] has a significant tyrosinase inhibitory activity, and its anti-tyrosinase activity is better than quercetin-3-*O*-*α*-L-rhamnoside (**11**) [26] and kaempferol (**7**) [26], which is consistent with the literature data. This indicated that 3′- and 4′-hydroxy groups on the B ring and 3-hydroxy group on the C ring in flavonols were crucial to their activities. The anti-tyrosinase activity of kaempferol (**7**) [26] is better than kaempferol-3-rhamnoside (**8**) [26], and compounds **8** and **12** [31] has no significant anti-tyrosinase activity, which further supports the above structure–activity relationships.

## 3. Experimental

### 3.1. Materials

The roots of *M. normale* were collected from Yanshan Town, Guilin City, Guangxi Province, in September 2018, and authenticated by Professor Yusong Huang (Guangxi Institute of Botany). A voucher specimen (20180912) was deposited in the Guangxi Key Laboratory of Functional Phytochemicals Research and Utilization, Guangxi Institute of Botany, China.

### 3.2. General Experimental Procedures

Optical rotations were measured with an ADP440+ polarimeter (λ 589 nm, path length 1.0 cm). The NMR spectra were acquired on a Bruker Advance 500 spectrometer (Bruker Biospin AG, Fällanden, Switzerland), and the residual solvent peaks were used as references. Coupling constants and chemical shifts were given in Hz and on a *δ* (ppm) scale, respectively. The ESIMS and HRESIMS data were recorded on a BRUKER HCT mass spectrometer and LCMS-IT-TOF spectrometer (Shimadzu, Kyoto, Japan), respectively. GC was performed on an Agilent 7890N (Agilent Technologies, Inc. Chandler, AZ, USA) system with a 0.32 mm i.d. × 25 m L-Chirasil-Val column. Analytical HPLC was run on a Shimadzu LC-2030C HPLC (Shimadzu, Kyoto, Japan) system using a 4.6 i.d. × 250 mm Agilent Eclipse XDB-C_18_ (5 μm) column. Semi-preparative HPLC was conducted on a Shimadzu LC-20AT HPLC system using a 20.0 i.d. × 250 mm Dr. Maisch reprosil 100 C18 (5 μm) column at a flow rate of 4 mL/min. Column chromatography (CC) was performed using Sephadex LH-20 (25–100 μm; GE Healthcare Bio-Science AB, Uppsala, Sweden), silica gel column (200–300 mesh, Qingdao Marine Chemical Co. Ltd., Qingdao, China), MCI gel CHP 20P (75–150 μm; Mitsubishi Chemical Co., Tokyo, Japan), and Chromatorex ODS (50 μm, Merck, Darmstadt, Germany) columns.

### 3.3. Extraction and Separation

Air-dried, powdered roots (10 kg) of *M. normale* were extracted with 80% aqueous acetone (3 × 7 days) at room temperature and each extract filtered. The filtrates were dried under reduced pressure to afford a crude extract (0.6 kg). The extract was suspended in H_2_O (1 L) and successively partitioned with petroleum ether (3 × 2 L), EtOAc (3 × 2 L). The EtOAc extract (80 g) was divided into ten fractions (Fr.1–10) by silica gel CC (8 i.d. × 20 cm) eluting with a gradient of CH_2_Cl_2_-MeOH (100:0, 95:5, 90:10, 80:20, 70:30, 50:50, 0:100, *v*/*v*). Fr.4 (32 g) was loaded onto an MCI gel column (6 i.d. × 20 cm) and eluted with a gradient of MeOH-H_2_O (0:100–100:0, *v*/*v*) to afford eighteen subfractions (Fr 4-1–4-18). Separation of subfraction Fr 4-14 (3.0 g) was done by another silica gel column (3 i.d. × 20 cm) eluting with a gradient of CH_2_Cl_2_-MeOH (97:3–80:20, *v*/*v*) and Sephadex LH-20 CC eluting with CH_2_Cl_2_-MeOH (1:1, *v*/*v*) to afford **7** (11.3 mg), **8** (10.6 mg), **9** (9.3 mg), **10** (20.6 mg), **11** (7.6 mg), and **12** (6.2 mg). Fr 6 (4.2 g) was applied to ODS C18 column and eluted with MeOH-H_2_O (20:80–80:20) to obtain 9 subfractions (Fr 6-1–6-9). Further separation of subfraction Fr 6-4 (1.2 g) using Sephadex LH-20 CC (eluted with MeOH-H_2_O, 10:90–100:0) yielded compounds **1** (9.6 mg), **5** (7.2 mg), and **6** (7.3 mg), respectively. Compounds **2** (*t*_R_ 130.5 min, 5.2 mg), **3** (*t*_R_ 146.8 min, 9.6 mg), and **4** (*t*_R_ 177.2 min, 15.3 mg) were obtained from Fr 7 (2.5 g) via semipreparative HPLC eluting with a gradient of MeOH-H_2_O (20:80–40:60, *v*/*v*, 0–250 min).

### 3.4. Spectroscopic Data

*Whiskey tannin C* (**1**): Brown amorphous powder; [α]${}_{D}^{25}$ −12.6° (*c* = 0.16, MeOH); UV (MeOH) λ_max_ nm (log *ε*): 203 (2.02), 276 (1.19); CD (MeOH) λ_max_ (Δ*ε*) 263 (−7.8), 240 (10.9), 225 (2.9). ^1^H and ^13^C NMR data, see Table 1; HRESIMS *m*/*z*: [M − H]^−^ 1005.1240 (calcd for C_45_H_33_O_27_^−^, 1005.1215).

*1-O-(4-methoxygalloyl)-6-O-galloyl-2,3-O-(S)-hexahydroxydiphenoyl-β-d-glucose* (**2**): Brown amorphous powder amorphous powder; [α]${}_{D}^{25}$ −5.5° (*c* = 0.17, MeOH); UV (MeOH) λ_max_ nm (log *ε*): 203 (2.52), 275 (2.01); CD (MeOH) λ_max_ (Δ*ε*) 264 (−8.0), 238 (30.6), 218 (1.3). ^1^H and ^13^C-NMR data, see Table 2; HRESIMS *m*/*z*: 799.0983 [M − H]^−^ (calcd for C_35_H_27_O_22_^−^, 799.0999).

*1-O-galloyl-6-O-(3-methoxygalloyl)-2,3-O-(S)-hexahydroxydiphenoyl-β-d-glucose* (**3**): Brown amorphous powder; [α]${}_{D}^{25}$ −33.5° (*c* = 0.12, MeOH); UV (MeOH) λ_max_ nm (log *ε*): 203 (2.40), 273 (1.68); CD (MeOH) λ_max_ (Δ*ε*) 262 (−3.1), 237 (11.1), 210 (−2.7). ^1^H and ^13^CNMR data, see Table 2; HRESIMS *m*/*z*: 799.1006 [M − H]^−^ (calcd for C_35_H_27_O_22_^−^, 799.0999).

*1-O-galloyl-6-O-vanilloyl-2,3-O-(S)-hexahydroxydiphenoyl-β-d-glucose* (**4**): Brown amorphous powder; [α]${}_{D}^{25}$ −20.6° (*c* = 0.15, MeOH); UV (MeOH) λ_max_ nm (log *ε*): 203 (2.42), 278 (1.86); CD (MeOH) λ_max_ (Δ*ε*) 263 (−12.1), 239 (42.5), 219 (0.9). ^1^H and ^13^CNMR data, see Table 2; HRESIMS *m*/*z*: 783.1032 [M − H]^−^ (calcd for C_35_H_27_O_21_^−^, 783.1050).

### 3.5. Acid Hydrolysis of **2**–**4**

Compound **2** (2 mg) was treated with 1 M HCl (5 mL) at 80 °C for 4 h, and then extracted with ethyl acetate (3 × 5 mL). The aqueous phase was dried under a stream of N_2_ to generate a neutral residue that was analysed using TLC (SiO_2_) with EtOAc-pyridine-EtOH-H_2_O (7:1:1:2) as solvent system. The *R*_f_ value of the neutral residue was the same as that of authentic D-glucose indicated the sugar component of 2 was glucose. The neutral residue and 5 mg L-cysteine methyl ester hydrochloride were dissolved successively in 3 mL of anhydrous pyridine and warmed at 80 °C for 1 h. After removal of the solvent by evaporation under reduced pressure, the reaction mixture was subsequently reacted with 0.6 mL of N-trimethylsilylimidazole at 80 °C for 1 h. The reaction mixture was partitioned with n-hexane and H_2_O, and then the n-hexane layer was analysed by a GC instrument. The injector and detector temperatures were set at 250 °C and 280 °C, respectively. The initial column temperature was held at 160 °C for 1 min, then increased to 280 °C at 5 °C/min. and held for 10 min. The authentic D-glucose and authentic L-glucose were silylated and analyzed in the same way, and their retention times were 19.09 and 19.25 min, respectively. The results of the GC analysis indicated that the sugar component of **2** was D-glucose (*t*_R_ 19.1 min). The sugar components of **3** and **4** were determined using the same procedure.

### 3.6. Anti-Inflammatory Activity

#### 3.6.1. NO Production by LPS-Stimulated RAW 264.7 Cells

The RAW 264.7 cells were cultivated in DMEM supplemented with 10% FBS at 37 °C in a humidified atmosphere of 5% CO_2_ for 24 h. Cells in 24-well plate (5 × 10^4^ cells/well) were treated with 200 ng/mL LPS and the test compounds. After 22 h, the media were collected, and the level of nitrite was measured using the Griess Reagent System (Promega, Madison, WI, USA).

#### 3.6.2. Cell Viability 

MTT assay was carried out to measure cytotoxicity of test samples on the RAW264.7 cell line. Cells which were in logarithmic growth phase were seeded in 96-well plate at the density of 5 × 10^4^–6 × 10^4^ cells/well and incubated at 37 °C with 5% CO_2_ for 24 h. Next, the cells were treated with 100 μL of culture medium at various concentrations of test samples for 24 h. The medium was discarded and 100 μL of FBS free medium containing MTT (1 mg/mL) was added to each well. After incubation in the incubator for 4 h, the supernatant was discarded and 100 μL of DMSO was added to each well to dissolve the formazan. The absorbance was measured at 570 nm using a microplate reader.
Cell viability (%) = (absorbance of sample/absorbance of control) × 100

### 3.7. Anti-Tyrosinase Activity

The anti-tyrosinase activities of the isolates were investigated according to the procedure described by Aoki et al. with slight modifications [21]. L-DOPA and kojic acid were used as substrate and positive control, respectively. Solution of L-DOPA at 5.0 mM, mushroom tyrosinase at 100 U/mL, as well as samples at different concentrations were prepared in phosphate buffer (pH 6.8). 20 µL of the sample solution and 10 µL of mushroom tyrosinase solution were mixed and pre-incubated at 37 °C for 10 min, then 40 µL of L-DOPA solution was added and incubated at 37 °C for 5 min. The reaction system of anti-tyrosinase activity experiment is shown in Table 4. The absorbance was measured at 475 nm using the Spark 10M multimode microplate reader (Tecan Trading AG, Zurich, Switzerland). All assays were repeated three times and each time in triplicate. The percent inhibition of tyrosinase activity was calculated using the following formula:Tyrosinase inhibition (%) = [1 − (T − T_0_)/(C − C_0_)] × 100 
where T represents the absorbance with sample and tyrosinase; T_0_ represents the absorbance with sample but no tyrosinase; C represents the absorbance with tyrosinase but no sample; C_0_ represents the absorbance without tyrosinase and sample.

## 4. Conclusions

The study is intended to explore more anti-inflammatory and enzyme inhibitory activities of polyphenols from the roots of *M. normale* on the basis of our previous works [20]. As expected, twelve polyphenols were obtained from the tile plant for the first time, and compounds **1**–**4** were new ellagitannin. The successful isolation and structure identification of ellagitannin provide materials for the screening of anti-inflammatory drugs and enzyme inhibitors, and also contribute to the development and utilization of *M. normale*. Ellagitannins **1**–**4** have no significant tyrosinase inhibitory activities and anti-inflammatory activities, which makes it less likely to be a potential anti-inflammatory drug or enzyme inhibitor. Fortunately, they possess new effects, such as anti-obesity [6,7]. Flavonoids have better anti-inflammatory and inhibitory enzyme activities than ellagitannins in the roots of *M. normale*, and they are more likely to be anti-inflammatory and anti-enzyme inhibitor. The study of structure–activity relationship is helpful to find new anti-inflammatory drugs and enzyme inhibitors.

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
