# Peer review of "Structural Characterization and Assessment of Anti-Inflammatory and Anti-Tyrosinase Activities of Polyphenols from Melastoma normale"

_molecules, 2021, doi:10.3390/molecules26133913_

Round 1
Reviewer 1 Report
- The authors report the isolation of four new compounds from Melastoma normale, which is excellent, but why did they choose to determine anti-tyrosinase activity, when the plant is used traditionally to treat dysentery, leukorrhea, and traumatic bleeding? This should be explained.
- The tyrosinase inhibition of quercetin and quercetin-3-O-α-L-rhamnoside, the two compounds the authors classify as being “strong” inhibitors, has been reported by other authors (see namely Kishore et al., 2018, doi:10.1021/acs.jnatprod.7b00564) and Fan et al., 2017 (doi:10.1016/j.foodres.2017.07.010). This should be mentioned in the discussion, which is practically inexistent and where a comparison with other results previously published concerning the known compounds should be carried out. The activity of the four novel compounds is not very high, but their novelty justifies the bioactivity determination. The novelty of the compounds is not duly emphasized by the authors, when it is what brings originality to this report. Have the other known compounds been assessed for this activity?
- Basically, the manuscript lacks a proper discussion and conclusions, comparing the results of the known compounds and emphasizing the novelty of what is new, so this should be rewritten. The Introduction must also be improved, namely by justifying the choice of bioactivity determined.
- Please correct this sentence: on line 273, in “Conclusions”, surely the authors wrote “cytotoxic” instead of “anti-tyrosinase” activities.
Reviewer 2 Report
The present work reports the results of structural characterization and assessment of the tyrosinase activity of polyphenols from Melastoma normale. I recommend this manuscript for publication in Molecules after improving certain aspects of the manuscript.
- Elements of scientific novelty should be presented in a detailed and convincing manner (in the last paragraph of the Introduction). In addition, it should also be briefly described in the Abstract.
- The Abstract should be re-written. It is not interesting at all and does not give the required information, which could be deducted from the main text.
- The current importance of the field should be clearly given in the Introduction.
- I suggest that a diagram (scheme) presenting the used procedures used in the study should be added to Experimental section.
- Somewhere, the results±SD are given in bad way (e.g. table 3). Two significant digits should be considered. Please, correct where required (tables and text).
- Innovative potential of the results obtained should be explained in detail (CONCLUSIONS).
- Little bits of minor English corrections required through the manuscript.
Reviewer 3 Report
The manuscript is interesting, the authors used a current methodology consistent with the objective of the study. However, I have the following comments.
I. Major Comments:
1. Improve the writing of the abstract (structure and writing).
2. In the introduction it is necessary:
2.1. The authors do not appropriately show the background that ustify the study (this will greatly limit the interest in the manuscript).
2.2. Include aspects related to the benefits of polyphenols.
3. The presentation of the results is good, but the discussion is very short. It is necessary to increase the discussion, especially the potential benefits and even clinical applications.
4. Separating the discussion from the results would be good.
5. Why study and identify these polyphenols? Will they be biomedical or have a potential clinical application? Discuss this point.
6. Polyphenols have very broad cytoprotective effects (antioxidants, anti-inflammatory, etc). These effects have been extensively studied. For example, protective effects in obesity models. It would be interesting to discuss this point.
Suggested references:
Hydroxytyrosol supplementation ameliorates the metabolic disturbances in white adipose tissue from mice fed a high-fat diet through recovery of transcription factors Nrf2, SREBP-1c, PPAR-γ and NF-κB. Biomed Pharmacother. 2019; 109: 2472-2481.
Molecular adaptations underlying the beneficial effects of hydroxytyrosol in the pathogenic alterations induced by a high-fat diet in mouse liver: PPAR-alpha and Nrf2 activation, and NF-kappaB down-regulation. Food Funct. 2017; 8: 1526-1537.
II. Minor comments:
1. Improve the writing of the study objective.
2. Some sentences are very long, I suggest checking the wording.
Round 2
Reviewer 2 Report
I accept this version
Reviewer 3 Report
Authors made all changes suggested. Manuscript can be accepted.